

# Clouds influence the functioning of airborne microorganisms

Raphaëlle Péguilhan[1†], Florent Rossi[1‡], Muriel Joly[1], Engy Nasr[2], Bérénice Batut[2ꟼ], François Enault[3], Barbara Ervens[1], Pierre Amato[1]*

[1] Université Clermont Auvergne, Clermont Auvergne INP, CNRS, Institut de Chimie de Clermont-Ferrand, F-63000 Clermont–Ferrand, France.

[2] Department of Computer Science, University of Freiburg; Freiburg, 79110, Germany.

[3] Université Clermont Auvergne, CNRS, Laboratoire Microorganismes : Génome et Environnement (LMGE), F-63000 Clermont-Ferrand, France.

[†] Now at: Department of Chemical and Biochemical Engineering, Technical University of Denmark, DK-2800 Kgs. Lyngby, Denmark.

[‡] Now at: Département de Biochimie, de Microbiologie et de Bio-informatique, Faculté des Sciences et de Génie, Université Laval ; Québec, Canada.

[ꟼ] Now at: Institut Français de Bioinformatique, CNRS UAR 3601, France & Mésocentre Clermont-Auvergne, Université Clermont Auvergne, Aubiere, France.

*Correspondence to*: Pierre Amato (pierre.amato@uca.fr)

**Abstract.** Airborne micro-organisms can remain at altitude for several days exposed to multiple environmental constraints that prevent or limit microbial activity, the most important of which is probably the lack of available liquid water. Clouds, *i.e.* air masses containing liquid water, could offer more favorable conditions. In order to investigate the influence of clouds on the functioning of airborne microorganisms, we captured aerosols into a nucleic acid preservation buffer from a high-altitude mountain meteorological station during cloudy and clear conditions, and examined metatranscriptomes. The specificities of aeromicrobiome's functioning in clouds compared to the clear atmosphere were then decrypted from differential functional expression analysis (DEA). The data reveal higher RNA-to-DNA content in clouds than in the clear atmosphere suggesting higher metabolic activity, and an overrepresentation of microbial transcripts related to energy metabolism, the processing of carbon and nitrogen compounds, intracellular signaling, metabolic regulations, transmembrane transports, and others. Stress response orients towards responses to osmotic shocks and starvation, rather than the defense against oxidants in clear atmosphere. Autophagy processes in eukaryotes, (macropexophagy, *i.e.* the recycling of peroxisomes) could help to alleviate the limited amounts of nutrients in the restricted microenvironments provided by cloud droplets. The whole phenomenon resembles the rapid resumption of microbial activity in dry soils after rewetting by rain, known as the "Birch effect", described here for the first time in the atmosphere. This work provides unprecedented information on the modulations of aeromicrobiome functioning in relation to atmospheric conditions. In addition of contributing to the processing and fate of chemical compounds in the atmosphere, cloud-induced modulations of biological processes could have ecological repercussions by shaping airborne microbial diversity and their capacity to invade surface environments.





## 1 Introduction

It is well established that biological material circulates in the atmosphere. This includes taxonomically and functionally diverse
microorganisms, with frequent saprotrophs (Amato et al., 2007b; Tignat-Perrier et al., 2020), methylotrophs (Amato et al., 2007a), phototrophs (Dillon et al., 2020) and others including obligatory or opportunist plant, animal or human pathogens (Brown and Hovmøller, 2002). Once aerosolized as individual cells or fragments of biofilms, they can remain airborne for up to several days (Burrows et al., 2009), at concentrations typically ranging from ~$10^3$ to ~$10^6$ cell $m_{(air)}^{-3}$ (Amato et al., 2023; Šantl-Temkiv et al., 2022). Close to the ground, in the planetary boundary layer, the airborne microbial diversity reflects that
of the emitting surfaces and follows its spatial and temporal variations in relation to meteorological and (micro)climatic conditions (Bowers et al., 2011; Fierer et al., 2008; Gusareva et al., 2019; Prass et al., 2021; Tignat-Perrier et al., 2020). At high altitudes in the free troposphere, the plumes from multiple sources mix which results in more evenly distributed assemblages (Péguilhan et al., 2021) at extremely low biomass (Smith et al., 2018).

Viable microbial cells suspended in the air are exposed to conditions at the limits of life's capacities including low water and
nutrient availability, low temperatures, and high levels of UV radiation and oxidants (Šantl-Temkiv et al., 2022). Water availability in particular is among the most limiting factors of biological processes in nature (Stevenson et al., 2015). Clouds, that occupy ~15% of the volume of the lower troposphere (Lelieveld and Crutzen, 1990), could thus act as "oases" within the otherwise vast and hostile atmospheric environment by offering liquid water and dissolved organic nutrients to surviving airborne cells.

Clouds are air volumes where relative humidity exceeds 100%, resulting in the condensation of water vapor on the surface of aerosol particles, including microbial cells. This leads to the formation of droplets of a few micrometers in diameter (*i.e.* individual volumes of ~$10^{-6}$ µL), with a typical liquid water content of ~0.1-1 $g.m_{(air)}^{-3}$. Chemical compounds from the gas and particle phases dissolve into the aqueous phase, and complex chemical processes take place with notable influence the composition of air masses (Ervens et al., 2018; Herrmann et al., 2015; Lelieveld and Crutzen, 1990; Li et al., 2023).
Microbiological processes as well can, to some extent, participate to process organic compounds and oxidants (Bianco et al., 2019; Khaled et al., 2021; Vaïtilingom et al., 2013). From the perspective of the microbiologist, cloud droplets can thus be considered short-lived aquatic microhabitats providing microorganisms with liquid water and a range of dissolved nutrients at nano- to micro-molar concentration (carboxylic acids, amino acids, ammonium, nitrate, metals, etc.) (Deguillaume et al., 2014; Šantl-Temkiv et al., 2013). Bulk cloud water was indeed evidenced during laboratory incubations to offer nutritional conditions
compatible with microbial development, with impacts on the chemical composition (Amato et al., 2007a; Bianco et al., 2019; Sattler et al., 2001; Vaïtilingom et al., 2013). In addition, clouds, through precipitation, provide efficient access routes to the ground for micro-organisms airborne at high altitude and thus contribute to aerial dissemination (Péguilhan et al., 2021; Woo and Yamamoto, 2020).

The highly diluted microbial biomass in the atmosphere, along with short residence time, make any in-situ assessment
challenging, so how the functioning of living cells may be modulated during atmospheric transport remains largely unexplored.



If conditions allow, airborne microorganisms can maintain or activate metabolic processes in response to environmental conditions (Amato et al., 2017; Hill et al., 2007; Klein et al., 2016; Šantl-Temkiv et al., 2018). For instance, bacteria (Sphingomonas aerolata) aerosolized in a simulation chamber increases ribosome numbers when exposed to volatile organic compounds (ethanol, acetic acid) and so, potentially, metabolic activity (Krumins et al., 2014). Data also suggest modulations
of the energy metabolism of living bacteria in natural clouds in relation with oxidants (Wirgot et al., 2017).

So far, current knowledge of microbial functioning in the atmosphere and clouds is thus based almost exclusively on laboratory incubations of samples and isolated strains (Amato et al., 2007a; Bianco et al., 2019; Jousse et al., 2018; Vaïtilingom et al., 2013; Wirgot et al., 2019) or, at best, on experiments in atmospheric simulation chambers (Amato et al., 2015; Krumins et al., 2014), *i.e.*, in conditions that do not fully reflect the in-situ natural atmospheric conditions in which the cells are actually
exposed. Metagenomics and, in particular, metatranscriptomics can provide instant snapshots of the biological processes taking place in a system. Over the last decade, the advent of high-throughput sequencing techniques stimulated such approaches. These led to unprecedented insights into the functioning of microbiota in humans (Franzosa et al., 2014; Jorth et al., 2014), oceans (Salazar et al., 2019), rivers (Satinsky et al., 2014), soils (Rosado-Porto et al., 2022), and highly polluted environments (Chen et al., 2015). Clouds were explored once, revealing multiple biological processes including responses to stresses,
transport and central catabolic and anabolic processes (Amato et al., 2019). By comparing cloud transcriptomes with other data available in the literature, this work highlighted functional peculiarities compared with surface biomes. Still, there is no information regarding possible specificities of microbial functioning in clouds compared to the clear, cloud-free atmosphere, which occupies most of the atmospheric volume. Here we intend to clarify this using a combined non-targeted metagenomics/metatranscriptomics approach.

## 2 Materials and methods

### 2.1 Sample collection

Samples were collected from the summit of Puy de Dôme Mountain (PUY; 1 465 m a.s.l., 45.772° N, 2.9655° E, France, ~400 km East from the Atlantic Ocean and ~300 km North of the Mediterranean Sea), located in an area composed of deciduous forests and pastoral landscapes and exposed most of the time to air masses from North and West (Deguillaume et al., 2014;
Renard et al., 2020). This mountain station is part of the Cézeaux-Aulnat-Opme-Puy-de-Dôme (CO-PDD) instrumented platform network for atmospheric research (Baray et al., 2020). Meteorological variables are monitored and the station is fully equipped for on-site sample processing and conditioning, including for microbiological and molecular analyses.

The main information pertaining to sample acquisition is summarized in Table 1. A total of nine cloud and six clear air events were sampled in 2019 and 2020, for periods of about two to six consecutive hours. In both conditions, two to four high-flow-
rate impingers (HFRi; model DS6, Kärcher SAS, Bonneuil-sur-Marne, France) sampling with an air-flow rate of 2 m³ min⁻¹ were deployed in parallel. More details about these samplers and their applicability to collect biological material are provided in (Šantl-Temkiv et al., 2017). Nucleic acid analyses were carried out from samplers filled with Nucleic Acid Preservation



(NAP) buffer solution (Camacho-Sanchez et al., 2013; Menke et al., 2017) as the collection liquid (1.7 L of 0.5X NAP for clear atmosphere, or 850 mL of 1X NAP for clouds, in order to account for expected liquid evaporation or accumulation),

following the procedures detailed in (Péguilhan et al., 2023a, b), including decontamination and controls. Briefly, the volume of the collection liquid was checked by weighting every sampling hour, and compensated if necessary with autoclaved ultrapure water. Samples were processed immediately after sampling using the PUY station's microbiology facility, within a laminar flow hood previously exposed to UV light for 15 min. The collection liquid from each individual sampler was filtered through 0.22 µm porosity mixed cellulose esters (MCE) filters (47 mm diameter; ref. 0421A00023; ClearLine®, Bernolsheim,

France) using sterile Nalgene filtration units. Filters were rolled using sterile forceps and placed into 5 mL Type A Bead-tubes (ref. 740799.50; Macherey-Nagel, Hoerdt, France). A volume of 1,200 µL of MR1 lysis buffer (ref. 744351.125; Macherey-Nagel) was then added to each tube, and a bead-beating step of 10 min was performed using a Genie2 vortex set at maximum speed. Filters and lysates were finally stored at -80°C in the bead tubes until further processing as detailed in the next section. Meteorological variables during sampling were monitored by the PUY meteorological station, including temperature, relative

humidity, liquid water content, wind speed and direction (https://www.opgc.fr/data-center/public/data/copdd/pdd). The planetary boundary layer height (BLH) was extracted from ECMWF ERA5 global reanalysis (https://www.ecmwf.int/en/forecasts/datasets/reanalysis-datasets/era5) (Hoffmann et al., 2019). The geographical origin of the sampled air masses was derived from 72-hour backward trajectories computed using the CAT trajectory model (Baray et al., 2020), which uses dynamical fields extracted from the ERA-5 meteorological data archive with a spatial resolution of 0.125°

for the present work. This tool was used for estimating percentages of air mass trajectory points in each of the eight direction sectors (Renard et al., 2020).



**Table 1. Conditions of sample acquisition.**

| SampleID | Sampling date (dd/mm/yyyy) | Sampling duration (h) | Geographical origin of the air mass[†] | Boundary layer height (min-max [average]) (m above sea level)[‡] | Position of the sampling site relative to the boundary layer | Temperature (°C)[§] | Relative humidity (%)[§] | Wind speed (m.s$^{-1}$)[§] | Liquid water content (LWC) (g.m$^{-3}$)[§] |
|---|---|---|---|---|---|---|---|---|---|
| **CLEAR CONDITIONS** | | | | | | | | | |
| 20200707AIR | 07/07/2020 | 6.5 | NW | 1268-1834 [1626] | In | 11.1 | 61 | 3.6 | < 0.01 |
| 20200708AIR | 08/07/2020 | 6.1 | NW | 623-1675 [1253] | In | 14.2 | 53 | 3.1 | < 0.01 |
| 20200709AIR | 09/07/2020 | 6.0 | N | 651-2377 [1487] | In | 20.3 | 48 | 3.4 | < 0.01 |
| 20200922AIR | 22/09/2020 | 5.9 | W | 665-1334 [972] | Out | 12.4 | 78 | 1.0 | < 0.01 |
| 20201118AIR | 18/11/2020 | 5.8 | W | 680-1142 [870] | Out | 14.1 | 41 | 6.4 | < 0.01 |
| 20201124AIR | 24/11/2020 | 6.0 | W | 644-740 [699] | Out | 8.6 | 50 | 3.4 | < 0.01 |
| **Minimum** | - | **5.8** | - | - | - | **8.6** | **41** | **1.0** | **< 0.01** |
| **Maximum** | - | **6.5** | - | - | - | **20.3** | **78** | **6.4** | **< 0.01** |
| **Median** | - | **6.0** | - | - | - | **13.3** | **52** | **3.4** | **< 0.01** |
| **Mean** | - | **6.1** | - | - | - | **13.5** | **55** | **3.5** | **< 0.01** |
| **Standard error** | - | **0.2** | - | - | - | **4.0** | **13** | **1.7** | **-** |
| **CLOUDS** | | | | | | | | | |
| 20191002CLOUD | 02/10/2019 | 2.4 | NW | 1422-1505 [1465] | In | 6.5 | 100 | 3.0 | NA |
| 20191022CLOUD | 22/10/2019 | 6.4 | S | 698-957 [813] | Out | 5.7 | 100 | 8.7 | NA |
| 20200311CLOUD | 11/03/2020 | 4.1 | W | 964-1145 [1060] | Out | 5.0 | 100 | 7.4 | NA |
| 20200717CLOUD | 17/07/2020 | 3.3 | NW | 1271-1437 [1343] | Out | 10.1 | 100 | 1.6 | 0.08 |
| 20201016CLOUD | 16/10/2020 | 4.7 | NE | 917-1034 [958] | Out | 1.1 | 100 | 1.8 | 0.35 |
| 20201028CLOUD | 28/10/2020 | 6.0 | W | 1026-1529 [1269] | Out | 5.2 | 100 | 11.0 | 0.23 |
| 20201103CLOUD | 03/11/2020 | 3.5 | W | 1126-1593 [1390] | In | 2.2 | 100 | 8.7 | 0.06 |
| 20201110CLOUD | 10/11/2020 | 3.1 | SW | 691-1276 [1016] | Out | 5.9 | 100 | 2.5 | 0.07 |
| 20201119CLOUD | 19/11/2020 | 2.8 | W | 1207-1234 [1215] | Out | 0.3 | 100 | 7.7 | 0.11 |
| **Minimum** | - | **2.4** | - | - | - | **0.3** | **100** | **1.6** | **0.06** |
| **Maximum** | - | **6.4** | - | - | - | **10.1** | **100** | **11.0** | **0.35** |
| **Median** | - | **3.5** | - | - | - | **5.2** | **100** | **7.4** | **0.10** |
| **Mean** | - | **4.0** | - | - | - | **4.7** | **100** | **5.8** | **0.15** |
| **Standard error** | - | **1.4** | - | - | - | **3.0** | **0** | **3.6** | **0.11** |
| **P-value** (Mann-Whitney test; clouds *vs* clear conditions) | | 0.04* | - | - | - | 0.003** | 0.001** | 0.44 | 0.003** |




## 2.2 Nucleic acid extraction and shotgun sequencing

For each sample, DNA and RNA were extracted in parallel from single MCE filters using NucleoMag® DNA/RNA Water kit (Macherey-Nagel, Hoerdt, France), following the protocols recommended by the manufacturer for filter membranes. All facilities were previously treated with RNase-away spray solution (Thermo Scientific; Waltham, USA). For DNA extraction,

half of the lysate was processed (600 µL), and a final step consisted of the removal of RNA by adding 1:50 volume of RNase A (12 mg.mL$^{-1}$, stock solution from Macherey-Nagel). DNA was finally eluted into 50 µL of DNase-free H2O after 5 min of incubation at 56°C, then quantified by fluorescence using the Quant-iT™ PicoGreen® dsDNA kit (Invitrogen; Thermo Fisher Scientific, Waltham, MA USA). For RNA extraction, the remaining 600 µL of lysate was processed, and the final step consisted of removal of DNA by the addition of 1:7 volumes of reconstructed rDNase (as provided with kits, Macherey-Nagel).

RNA was finally eluted into 30 µL of RNase-free H$_2$O after 10 min of incubation at room temperature. The total DNA and RNA concentrations in the air volumes sampled, as inferred from concentrations in the extracts, ranged between 0.12 and 2.19 ng.m$^{-3}$ and 0.06 and 1.27 ng.m$^{-3}$, respectively (Table S1). Individual DNA or RNA extracts from individual samplers from the same sampling event were pooled, and 30 µL were transferred to GenoScreen (Lille, France) for further processing of RNAs (quantification, reverse-transcription to cDNAs), and shotgun sequencing of the metagenomes (MGs) from DNAs, and

metatranscriptomes (MTs) from cDNAs, on Illumina HiSeq (paired end reads of 150 bp). A first sample (20191022CLOUD) was deeply sequenced (~200 M reads) and used to check the feasibility of the approach, adjust the sequencing depth, and elaborate bioinformatics workflows. The other samples were sequenced at a lower sequencing depth (40-60 M reads per sample). Raw sequencing MGs and MTs data are available as fastq.gz files through the European Nucleotide Archive at EBI, under the project accession PRJEB54740, samples ERR9966616 to ERR9966643.

## 140 2.3 Bioinformatics and differential expression analyses

Raw MGs contained approximately between 30 and 260 million (M) reads (68.7 M in average), and raw MTs from 65 M to 195 M reads (Tables S2-S3). The bioinformatics workflow is detailed in Supplementary Material and summarized in Fig. S1. Briefly, this consisted of (i) sequence preprocessing (quality control, trimming, etc.), (ii) taxonomic annotations of MGs and MTs [Kraken2 v2.1.1 (Wood and Salzberg, 2014) and "PlusPF" database], (iii) construction of a gene catalog to serve as a

unique reference for the study, as inspired from (Salazar et al., 2019) (Fig. S2). This was elaborated by (I) merging all the contigs from each individual MG, (II) predicting genes [MetaGeneAnnotator v1.0.0 (Noguchi et al., 2008)], (III) clustering in order to remove redundancy [CD-Hit v4.8.1 (Li and Godzik, 2006)], and (IV) annotating functions and taxonomy [using DIAMOND v2.0.8.0 (Buchfink et al., 2015) and the UniProtKB Swiss-Prot database (The UniProt Consortium, 2019)]. Finally, (iv) genes and transcripts in each MG and MT were mapped toward the annotated gene catalog. The log ratios of the

number of reads associated with a gene, taxon, E.C. or GO in a MT dataset to that in the corresponding MG (abbreviated as RNA:DNA log ratios) were calculated using data normalized to total counts. RNA:DNA ratios are commonly used as an appraisal of the relative level of metabolic activity, with higher ratios indicating potentially higher metabolic activity (Baldrian



et al., 2012; Zhang et al., 2014). In addition, statistical differential expression analysis was performed [DEA, MTX model v1.5.1 (Zhang et al., 2021)] to detect overrepresented genes and functions in MTs compared to MGs, and those overrepresented

in clouds compared to clear conditions or conversely. Metabolic pathways were reconstructed from the E.C. numbers obtained from UniprotKB identifiers using KEGG database and resources (Kanehisa et al., 2023).

## 3 Results

### 3.1 Microbial taxonomy in metagenomes and metatranscriptomes

The datasets include sequences from eukaryotes, bacteria, archaea and viruses. Bacteria dominate in clear atmosphere (~88%
and 71%, on average, of the total number of reads in MGs and MTs, respectively), while eukaryotes (mainly fungi) prevail in clouds (~51% and 87% on average, respectively) (Fig. S3), but both Prokaryotic and Eukaryotic diversity indices are statistically similar between cloud and clear atmosphere samples (Kruskal-Wallis tests, p > 0.05) (Fig. S4-S5). Archaea (Euryarchaeota, Thaumarchaeota and Crenarchaeota) and viruses (mainly bacteriophages) both contribute very low proportions of sequences (< 0.1%).

In Bacteria, a total of 32 distinct phyla, 159 orders and 1 249 genera are identified in MGs (Data S1). The dominant are Micrococcales, Corynebacteriales, Propionibacteriales (Actinobacteria), Pseudomonadales, Sphingomonadales, Burkholderiales and Hyphomicrobiales (Proteobacteria). The observed bacteria richness in samples varies between 532 and 826 genera, the vast majority of which (963 out of 1,249 genera; ~77%) are common to clouds and clear air. Shannon's diversity index ranges from ~2.6 to ~4.3 depending on samples (Fig. S4).

In Eukarya, identified richness distributes among 8 phyla, 21 orders and 54 genera (Data S2); all are shared between cloudy and clear atmospheric conditions (Fig. S5). Fungal taxa largely predominate, with the most abundant affiliated to Helotiales, Hypocreales and Mycosphaerellales in Ascomycota, and Ustilaginales in Basidiomycota. Other unicellular eukaryotic phyla detected include Apicomplexa (parasites of Metazoa), Bacillariophyta (microalgaes), Euglenozoa, Evosea, and Cercozoa (amoeboids and flagellates). The observed eukaryotic richness in samples varies between 41 and 43 genera depending on

samples, with Shannon's index ranging from ~1.1 to ~2.4 (Fig. S5).

The taxa significantly overrepresented in MTs compared with MGs include 77 families of bacteria (198 genera) and 10 families of eukaryotes (27 genera) (Data S3). These are subsets of the total biodiversity seen in MGs, and they are not distinguishable between clouds and clear conditions (PCA, Fig. S6). Bacterial taxa tend to exhibit higher relative representation in MTs compared to MGs (termed as RNA-to-DNA ratio) in clouds than in clear atmosphere, contrary to eukaryotes (Fig. S7-S8).

### 3.2 Functional aspects and differential expression analyses

The RNA-to-DNA concentration ratio is significantly higher in clouds than in a clear atmosphere (range 1.49-3.62 and 0.21-1.64, Mann-Whitney test, p-value=0.004) (Table S1; Fig. 1A). No relation is observed with relative humidity in clear atmosphere.




Independently from atmospheric conditions, abundant transcripts in the datasets are related to central biological functions and
their regulation, including carbon, amino-acid and protein processing, energy production, signalling, response to stresses,
transports and others (Fig. 2; panels A, C, E in Fig. 3 and Fig. S9-S10). From a total of 21,046 unique genes (Uniprot IDs) in
MGs constituting the reference catalog, 488 are found significantly more represented in MTs than in MGs (Data S4). These
correspond to, at the GO term level, 419 Biological Processes, 284 Molecular Functions, and 140 Cellular Components
overrepresented in transcriptomes (Data S5). Most of these (~80%) are affiliated with Eukaryotes, in particular Fungi, or with
Gamma-Proteobacteria and Actinobacteria in Bacteria (~48% and ~25% of the bacteria transcripts, respectively) (Fig. S11).

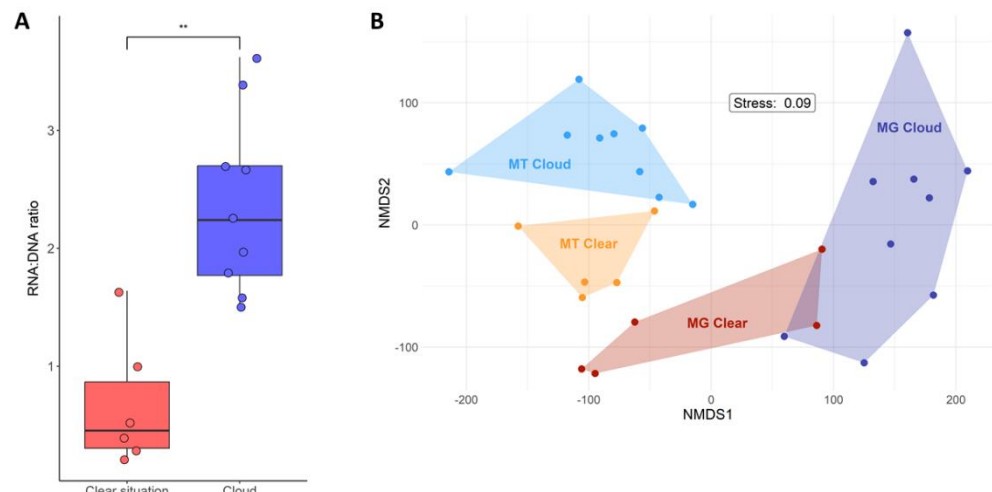

**Figure 1. (A) RNA-to-DNA concentration ratio in clouds and clear atmosphere; dots indicate individual samples, boxplots display
medians, 25th and 75th percentiles, and whiskers are 1.5 interquartile ranges; \*\* indicates a significantly higher ration in clouds
(Mann-Whitney test, p-value=0.004); (B) Non-Metric Multidimensional Scaling (NMDS) analysis based on the 21 046 functional
gene entries detected in total, depicting clear distinctions between MGs and MTs, and between MTs of cloudy and clear conditions.**

These genes and functions occur and distribute differentially depending on the presence of condensed water (panels B, D, F in
Fig. 3 and Fig. S10-S11). Multivariate analysis (NMDS) indeed indicates distinct transcriptional patterns depending on
atmospheric conditions (Fig. 1B), and differential expression analysis (DEA) specifies it (Fig. 4; Data S6-S7): among the 488
genes significantly more represented in MTs than in MGs, 320 (~66%) are also significantly differentially represented between
clouds and clear conditions, about two thirds of which in clouds, contributed by Eukaryotes (Fig. S12). In total, differentially
represented transcripts belong to 394 Biological Processes, 147 Cellular Components, 279 Molecular Functions, and
correspond to 200 unique E.C. numbers (see distribution of subclasses in Table S4). Overall, the most diverse enzyme
transcripts are NADH:ubiquinone oxidoreductase (E.C. 7.1.1.2, 11 distinct entries), RNA polymerase (E.C. 2.7.7.6), RNA
helicase (E.C. 3.6.4.13) and cytochrome-c oxidase (E.C. 7.1.1.9; 6 distinct entries each), and non-specific serine/threonine
protein kinase (E.C. 2.7.11.1; 5 distinct entries). We examined below the similarities and specificities of microbial



transcriptomes in and outside of clouds for different categories of functions and metabolisms based on the distribution of GOs and E.C. numbers (Fig. 4-5; Fig. S13-S17; Table S4; Data S6-S7).

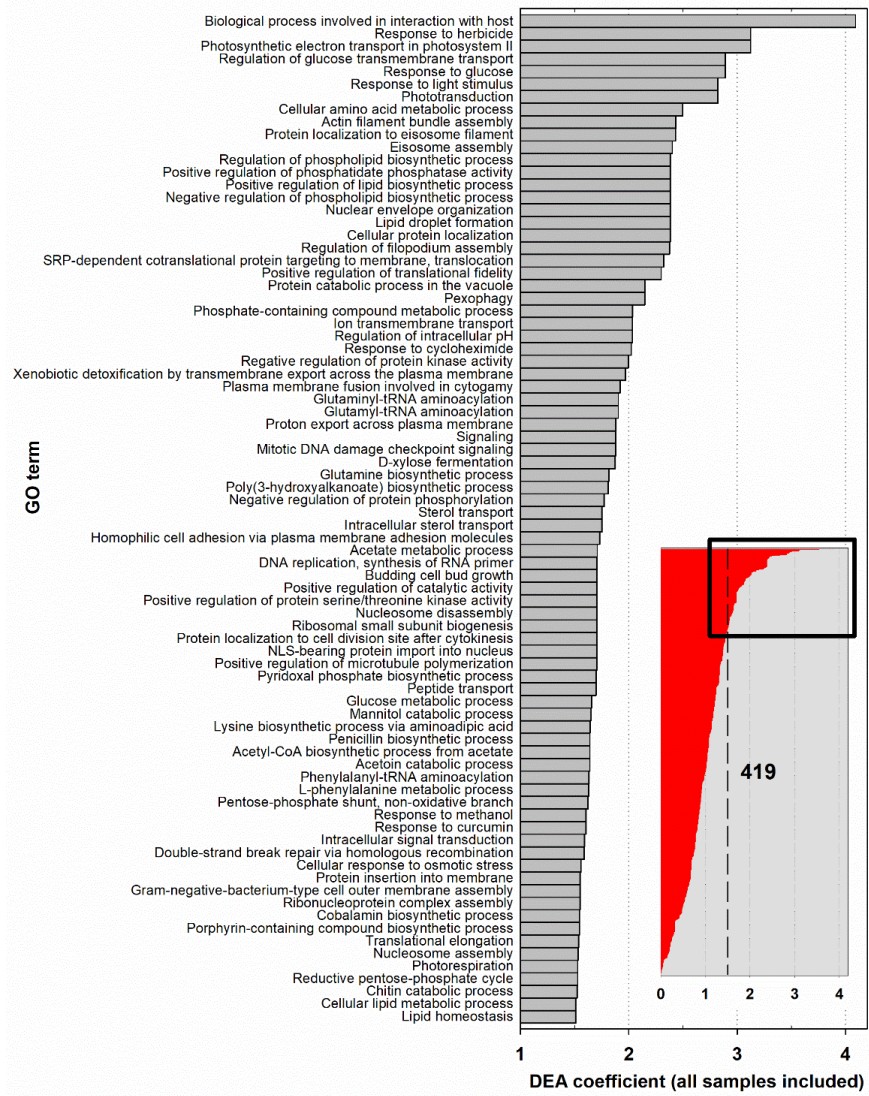

**Figure 2. Biological Processes GO terms associated with overrepresented transcripts in samples, as compared with their representation in metagenomes, from Differential Expression Analysis (DEA). Only the 80 GO terms with DEA coefficients > 1.5, out of 419 in total, are shown.**





**Figure 3. GO terms representation in genes and transcripts in metagenomes and metatranscriptomes (A; C; E), and (B; D; F) relative representation (termed as RNA:DNA) in metatranscriptomes in clouds *versus* clear conditions as compared with the corresponding metagenomes, for Biological Processes related to: central metabolism (A, B), energy metabolism (C, D), and catabolic processes (E, F); clr: centered log-ratio transformation. Other GO terms of interest are presented as Fig. S9-S10.**




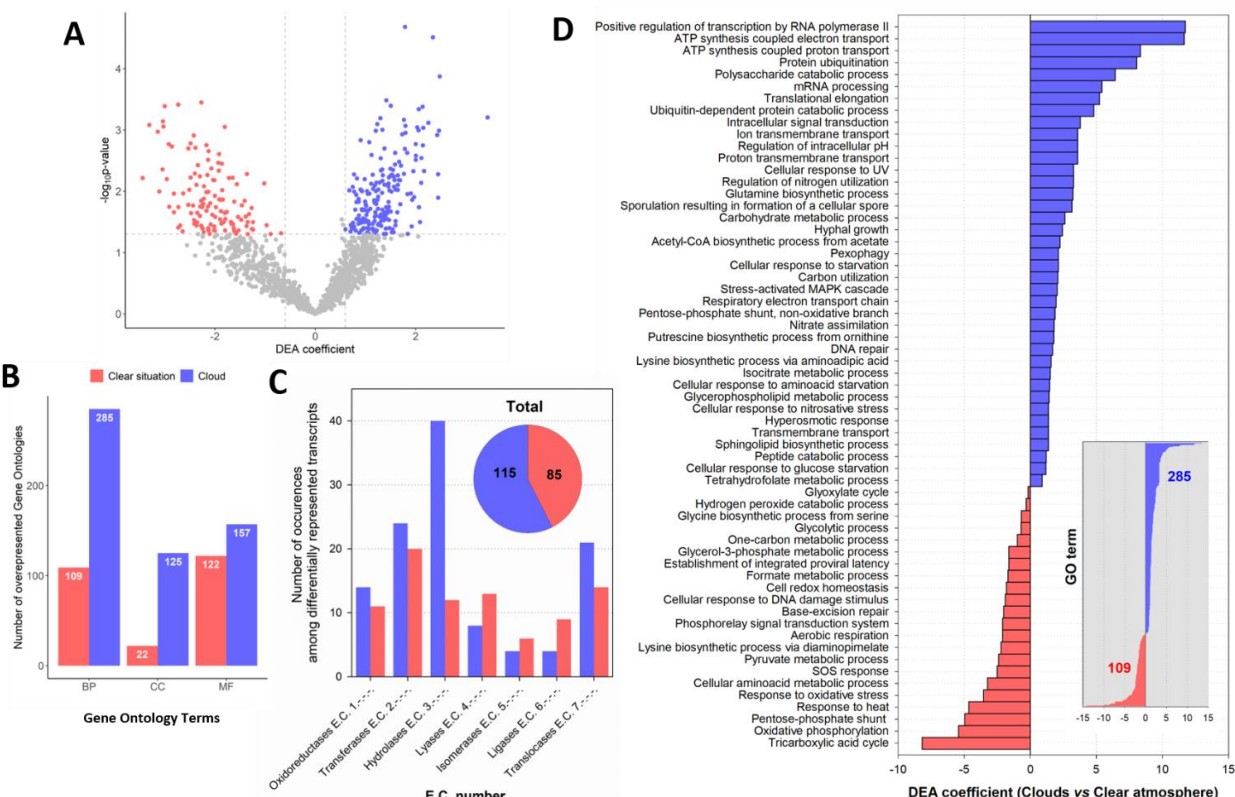

**Figure 4. Results of Differential Expression Analysis (DEA) of transcripts overrepresentation in clouds (blue) compared with clear atmosphere (red), at the gene (A), GO term (BP: Biological Processes; CC: Cellular Component; MF: Molecular Function) (B) and E.C. number (C) levels. In (D) the DEA coefficient associated with selected Biological Processes GO terms are shown (39 GO terms out of 285 in total for clouds and 22 out of 109 for clear atmosphere).**



### 3.2.1 Central, carbon and energy metabolisms

Large numbers of transcripts relate to central functions of carbon and energy metabolisms in both clouds and clear air samples (Fig. 3, Fig. S9-S10): glycolysis and glucose-related metabolic processes (GO:0006096; GO:0006006), glyoxylate and TCA cycles (GO:0006097; GO:0006099), and carbohydrate metabolisms (GO:0005975; GO:0006083; GO:0019427). Consistently, transcripts coding for key enzymes of these pathways are abundant in both prokaryotes and eukaryotes, such as isocitrate dehydrogenase (IDH) (EC 1.1.1.42 and EC 1.1.1.41), glyceraldehyde-3-phosphate dehydrogenase (GAPDH) (EC 1.2.1.12) and aconitase (EC 4.2.1.3) (Fig. 5, Fig. S13-S15).

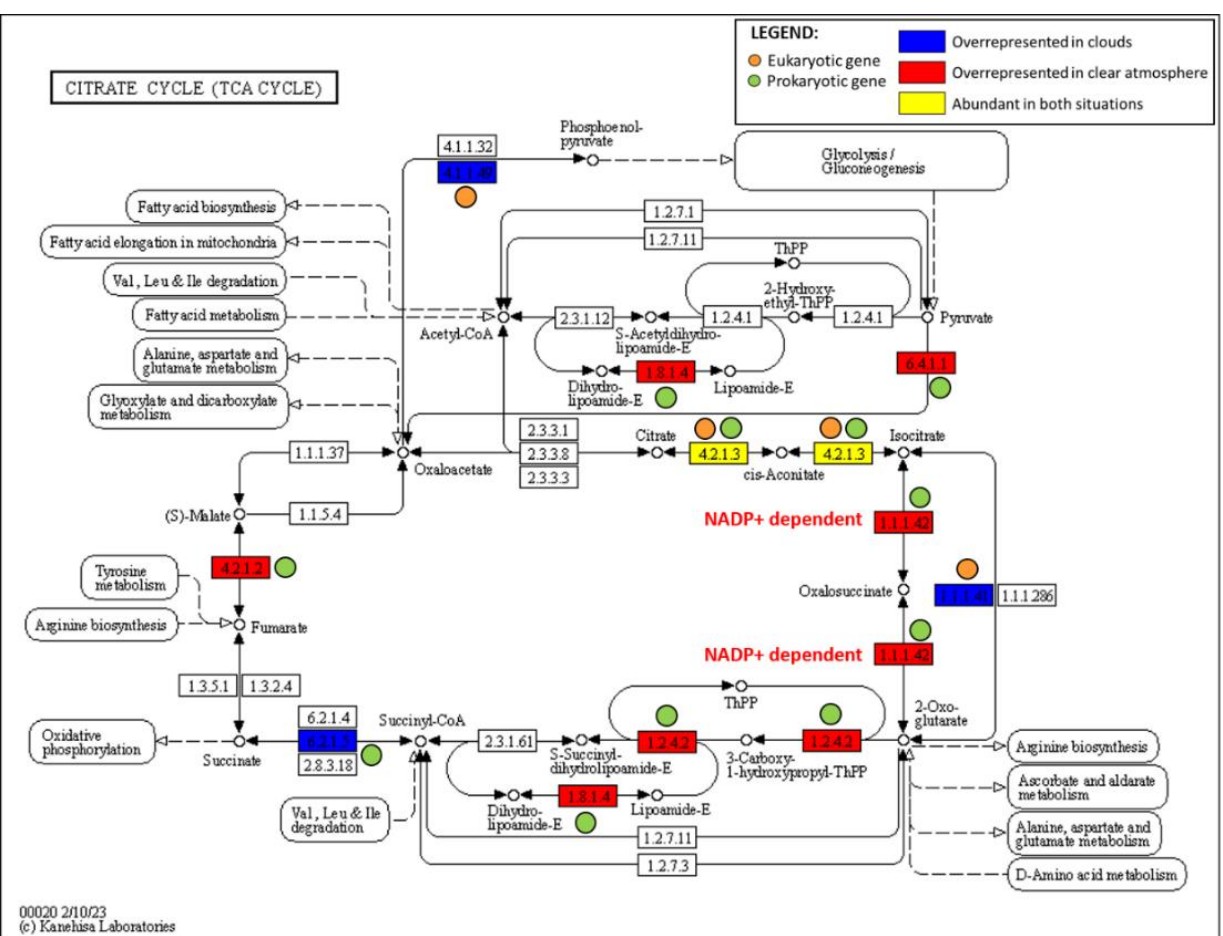

**Figure 5. The TCA cycle metabolic pathway, depicting overrepresented enzyme transcripts in clouds and/or in clear atmosphere by eukaryotes and/or by prokaryotes (from UniprotKB identifiers and KEGG database).**

DEA analysis indicates a strong overrepresentation in clouds of functional transcripts coding for hydrolases (E.C. 3.-.-.-) and, to a lesser extent, translocases (E.C. 7.-.-.-), particularly those involved in the translocation of protons (E.C. 7.1.-.-). These relate to GOs including carbon utilization (GO:0015976), polysaccharide catabolism (GO:0000272), and ATP production-coupled electron and proton transport (GO:0042773; GO:0015986) (Fig. 4). The pentose phosphate pathway (non-oxidative





phase) (GO:0009051) also tends to be more represented in clouds than clear atmosphere, with an over-representation in particular of fructose-bisphosphate aldolase transcripts (*fba*; EC 4.1.2.13) (Fig. S14).

In turn, transcripts of lyases (E.C. 4.-.-.-) and ligases (E.C. 6.-.-.-) are overall more prevalent in clear atmosphere than in clouds (Fig. 4; Table S4). In particular the TCA cycle pathway (GO:0006099) appears upregulated by prokaryotes, with transcripts coding for enzymes including pyruvate carboxylase (*pyc*, *pycA*, *ylaP*; E.C. 6.4.1.1), isocitrate dehydrogenase [NADP-
dependent] (IDH) (*icd*, *Cgl0664*, *cg0766*; EC 1.1.1.42), fumarate hydratase (*fumC*; E.C. 4.2.1.2), and alpha-ketoglutarate dehydrogenase (*sucA*, *odhA*, *Oant*; EC 1.2.4.2) (Fig. 5). Glycolytic processes and the glyoxylate cycle are overall barely affected by clouds based on DEA. Nevertheless, transcripts of enzymes involved in specific steps of these pathways are detected in higher proportions in clouds, including enolase (2-phospho-D-glycerate hydro-lyase; EC 4.2.1.11), phosphoenolpyruvate carboxykinase (EC 4.1.1.49), and acetyl-coenzyme A synthetase (E.C. 6.2.1.1) (Fig. S13-S14).

**3.2.2 Protein, amino-acids and nitrogen metabolism**

Numerous of the most abundant transcripts related to protein, amino acids and nitrogen metabolisms are more prevalent in clouds than they are in clear atmosphere (Fig. 3-4; Fig. S9-S10). These included post-translational protein modifications (ubiquination, phosphorylation and glycosylation, and related processes; GO:0016567; GO:0006511; etc.), known to participate to the regulation of enzymatic activities and multiple other cellular and metabolic processes (Yang et al., 2022).
Clouds are also associated with aminoacid starvation (GO:0034198). Consistently, among aminoacid related transcripts and functions, only glutamine and lysine biosynthesis are overrepresented in clouds (Fig. 4). Catabolism is rather directed towards serine and arginine, in link with ornithine and putrescine production pathways within glutathione metabolism, a major pathway in the recycling of NADP+ and the regulation of oxidants (Fig. S16). During clear conditions, processes of aminoacid biosynthesis prevail (arginine, leucine, threonine, lysine, beta-alanine, glycine), potentially involving gaseous dinitrogen
fixation (Dalton and Kramer, 2006).

Among inorganic nitrogen metabolism, nitrate assimilation (GO:0042128) and the utilization of ammonium NH4+ (GO:0042128) are both overrepresented in clouds. Both ions are abundant in atmospheric water (Péguilhan et al., 2021). The latter function concords with the overrepresentation of L-glutamate:ammonia ligase transcripts (E.C. 6.3.1.2; GO:0004356, gln), an enzyme involved in the synthesis of glutamine, one of the main regulators of cellular development and oxidative stress
response in fungi (Wang et al., 2022).

**3.2.3 Transport, signalling and response to stress**

In clouds, transcripts in link with transmembrane transports of ions and protons (GO:0034220, GO:1902600, GO:0055085) are among the most represented (Fig. S10). These participate in ATP synthesis (GO:0042773, GO:0015986 etc), the regulation of substrate utilization (GO:0006808, GO:0015976; GO:0009267) and homeostasis (GO:0045454) (Fig. 4; Fig. 6).



Oxidative stress (GO:0006979) and SOS response (GO:0009432) dominate in clear atmosphere, whereas in clouds responses to osmotic and nitrosative stress (GO:0071470; GO:0071500), stress-activated MAPK cascade (GO:0051403), the regulation of intracellular pH (GO:0051453), processes of starvation towards carbon and nitrogen (GO:0042149, GO:0034198), and processes of autophagy and pexophagy (*i.e.*, macropexophagy; GO:0051403) prevail (Fig. 4; Fig. 6; Fig. S10).

**Figure 6. Networks linking Biological Processes (GO terms) related with stress responses, and ATP synthesis and ion transport. The colour scale represents associated DEA coefficients, with negative values (red shades) indicating a significant overrepresentation of related transcripts in clear conditions, and positive values (blue shades) in clouds. Node size is scaled to DEA coefficient absolute value. Arrows indicate relationships between GOs ("is a" or "part of" as specified).**





## 4 Discussion

We report here the most unique and comprehensive dataset to date regarding aeromicrobiome functioning in the natural environment. For the very first time, we demonstrate that atmospheric conditions influence multiple facets of microbial functioning in the natural atmosphere. In previous work, using targeted and untargeted transcriptomics approaches, we identified metabolically active bacteria and fungi in clouds (Amato et al., 2017), then depicted their functioning through the analysis of transcriptomes (Amato et al., 2019). Here, we extend knowledge by integrating non-cloudy atmosphere in our

analysis, so replacing clouds into the atmospheric context, *i.e.* as volumes embedded within clear atmosphere.

In terms of taxonomy, the airborne microbial assemblages in clouds are not distinguishable from those in clear atmosphere, as we recently reported from amplicons for bacteria (Péguilhan et al., 2023a). They consist of diverse Eukarya, Bacteria, Archaea and viruses, with Proteobacteria and Actinobacteria dominating in Bacteria, and Ascomycota and Basidiomycota in Eukaryotes, which is typical over continental vegetated areas [*e.g.*, (Bowers et al., 2013; Tignat-Perrier et al., 2020)]. Such a

similarity was to be expected given the large size of microbial aerosols and so their ability to act as cloud condensation nuclei, and considering that cell multiplication in clouds is unlikely to significantly affect the structure of microbial assemblages due to limited residence time (Ervens and Amato, 2020).

### 4.1 A "Birch effect" up in the sky

Water limitation is a great challenge that many microorganisms face in their natural habitats. Here the presence of liquid water

in clouds is associated with consistent transcriptional patterns, and we found no relation between biological processes and relative humidity in clear conditions. Clear air was collected at relative humidity between 41%-78%, *i.e.* below the deliquescence point of most aerosol particles (Cruz and Pandis, 2000; Li et al., 2014), and at the limits of compatibility with biological processes (Stevenson et al., 2015).

Higher RNA:DNA ratio in clouds than in cloud-free air suggests higher levels of microbial metabolic activity (Baldrian et al.,

2012; Salazar et al., 2019), which remains to be assessed quantitatively. Accordingly, transcripts related to multiple biological processes are overrepresented in clouds as compared with clear atmosphere, such as energy metabolism, carbohydrate and polysaccharide catabolism, transcription, translation, transmembrane transports, and metabolic regulation mechanisms. Homeostasis regulation, starvation and autophagy supplant the oxidative stress response and DNA repair functions (SOS response) that prevail during clear conditions. Our observations therefore suggest a phenomenon in clouds similar to the "Birch

effect" that occurs in dry soils in response to rewetting by rain, where the sudden influx of water triggers a burst of microbial activity (Griffiths and Birch, 1961; Unger et al., 2010). The "Birch Effect" in soils typically lasts for a few days, hence, much longer than the lifetime of clouds and individual cloud droplets (Feingold et al., 1996). The metabolic modulations described here should therefore apply to any cloud regardless of the time elapsed since its formation. As with soils – but necessarily to a much smaller extent because of the low biomass – the resurgence of microbial activity in clouds may lead to the release of



gaseous biogenic compounds such as $N_2O$ through aerobic ammonium oxidation, i.e. nitrification (Jørgensen et al., 1998), which is also supported by the overrepresentation of ammonium utilization process.

Considering samples from higher altitude above ground and investigating the transitions between clear and cloudy conditions through time series would help to specify temporal aspects of the responsiveness of airborne microbial assemblages to changing conditions.

Dry-wet alternance in ecosystems contributes to shaping microbial assemblages, activity and responsiveness to changing conditions. The lag-time after which microorganisms start recovering in soil upon rewetting shortens after repeated dry-wet cycles, due to selection of the most responsive ones (Zhou et al., 2016). Transcriptionally active taxa in our observations represent ~20% of the richness, and this could attest and result from such selection processes during aerial transport. Typical atmospheric residence times of microbial cells are on the order of a few days (Burrows et al., 2009), during which they may

undergo ~10 water evaporation-condensation cycles before precipitating (Pruppacher and Jaenicke, 1995). Whereas clouds can last for several hours (Dagan et al., 2018), individual cloud droplets can form and evaporate within a few minutes (Feingold et al., 1996). Such rapid fluctuations of water availability could favour the most responsive microorganisms such as Proteobacteria and Actinobacteria, which include numerous generalists with high metabolic flexibility (Chen et al., 2021), and which in fact often dominate airborne microbial assemblages (Amato et al., 2017; Péguilhan et al., 2021; Šantl-Temkiv et al.,

100  2022).

### 4.2 Airborne fungal spores initiate germination in clouds

It is likely that fungal spores initiate germination in clouds. These are propagules designed for (aerial) dispersal (Brown and Hovmøller, 2002), which germinate (i.e. initiate growth) when they reach favourable conditions of water availability. During germination, functions of cell protection give way to anabolic processes within minutes (van Leeuwen et al., 2013; Leroch et

al., 2013). Cellular growth and respiration are promoted by the sudden availability of water, associated with the release and solubilization of readily bioavailable organic compounds. While dormant, functions of cell protection prevail in spores, against osmotic stress, heat, and oxidants. In the presence of water, mitogen-activated protein kinase (MAPK) cascade signalling pathways mediate the activation of central metabolic functions of energy production and biosynthesis (van Leeuwen et al., 2013). Autophagy processes in particular can participate to the early steps of appressorium synthesis in parasitic and symbiotic

fungi (Veses et al., 2008), a structure designed to invade host cells. Although we did not consider time series, our observations therefore strongly concur with such a sequence, and it is reasonable to assert that airborne fungal spore germination occurs in clouds.

### 4.3 Biomass production potentially occurs

A number of biosynthetic processes including lipid, amino-acids and others, indicate potential biomass production in both

clouds and clear atmosphere. For instance, lysine biosynthesis, a pathway involved in peptidoglycan synthesis in bacteria (Gillner et al., 2013; Kobashi et al., 1999), occurs from aminoadipic acid in clouds and from diaminopimelic acid in clear

 

atmosphere. These might be responses to changing conditions including osmotic and temperature variations (Amato, 2013), and could also indicate cell growth and proliferation (Gally et al., 1993; Zhu and Thompson, 2019). Although not evaluable here, it is conceivable that bacterial multiplication occurs in clouds, and also possibly at high relative humidity (>75%) where

deliquescent aerosols can occur (Cruz and Pandis, 2000). Observations of fog have indeed suggested so (Fuzzi et al., 1997), and microbial growth has been evidenced in bulk cloud water incubated in the laboratory (Amato et al., 2007a; Bianco et al., 2019). In the high atmosphere, cell multiplication is necessarily constrained by short residence time and low temperatures, and it should therefore not be significant (Ervens and Amato, 2020).

## 4.4 Utilization of nutrients and interactions with chemistry

Cloud water contains various dissolved carbon and nitrogen compounds that can be used as nutrients by microorganisms: carboxylic acids, aldehydes, sugars, amino-acids, ammonium, nitrate, etc. (Amato et al., 2007a; Bianco et al., 2016, 2018; Deguillaume et al., 2014; Renard et al., 2022). The overrepresentation of transcripts related to carbon, ammonium and nitrate utilization in clouds in our dataset suggests the uptake of such dissolved compounds by microorganisms, as suggested earlier (Amato et al., 2007a, 2019; Jaber et al., 2021; Vaïtilingom et al., 2010). Despite the low biomass, the impact of microbial

activity on organic carbon chemistry has been assessed as potentially significant, depending on the volatility and solubility of the compounds (Ervens and Amato, 2020; Khaled et al., 2021; Nuñez López et al., 2024). The impacts on nitrogen species have been barely examined (Hill et al., 2007; Jaber et al., 2021). Estimates indicate potential biodegradation of amino-acids, but there is as yet no information on inorganic compounds such as ammonium and nitrate, which are among the most abundant ions in cloud water (Deguillaume et al., 2014).

Functions related to responses to starvation toward organic carbon and amino-acids in clouds indicate that nutrient requirements are actually not fully satisfied. In natural clouds, statistically only 1 out of ~10 000 droplets contains a microbial cell that potentially causes a very efficient and rapid depletion of nutrients in these small, independent volumes (Khaled et al., 2021) (~$10^{-6}$ µl for 20 µm diameter droplets, so a cell concentration of ~$10^9$ cells mL$^{-1}$ in biotic droplets). Autophagy processes may contribute to alleviating the nutritive needs. Peroxisomes, organelles dedicated to the detoxification of oxidants in

eukaryotes, are targeted in particular by autophagy (pexophagy). Their elimination could compromise the chances of survival if the cloud evaporates and the cells are again exposed to high levels of oxidative damage.

## 5 Concluding remarks and perspectives

The recovery of active metabolic processes in airborne microorganisms prior to their deposition could facilitate surface or host invasion if cloud precipitates. In turn, in the likely event where the cloud evaporates instead of precipitating, triggering

germination and sacrificing essential cellular structures while conditions may soon become inhospitable could compromise future chances of survival, and so of further dispersion. It thus remains to be evaluated whether these metabolic regulations offer an ecological advantage to microorganisms in such transient environments as clouds where they have no chance to

establish due to limited residence time, or whether they may influence survival in the atmosphere and invasion processes upon deposition.

Just as the presence of liquid water in clouds or the chemical composition of air in atmospheric chamber (Krumins et al., 2014) are associated with variations in the transcriptomes of airborne microbes, it is likely that other environmental variables play roles, such as day/night variations, temperature, or repeated osmotic shocks. These still need to be evaluated to gain a better understanding of the constraints imposed on air-dispersed microorganisms.

Metatranscriptomics approaches are undoubtedly among the most powerful methods to examine microbial functioning. Still,
they remain limited by multiple constraints inherently associated with them. For instance, they do not allow attributing biological processes to individual cells or to specific taxa with certainty. They are also limited by current functional genomic knowledge and existing databases, in particular in natural environments like the atmosphere where rare taxa often represent an important fraction of the richness (Péguilhan et al., 2023b). In the atmosphere, micro-organisms have a short residence time of a few days, and air volumes contain mixed populations of cells of different origins and atmospheric ages. Such analyses as
metatranscriptomics, based on RNA, assume a high turnover of these molecules in cells, in particular mRNA, and are therefore considered to reflect quasi-instantaneous cell activity. Nevertheless, ribosomes (rRNA) can persist several days at low temperatures (Schostag et al., 2020) and may therefore be the result of recent past activity. These "residual" signatures of activity, along with the high level of mixing, can blur our vision of the actual situation in such highly dynamic environment as the atmosphere. Finally, transcriptomes attest of potential cellular activity, but they do not provide quantitative information of
microbial activity in terms of fluxes of elements or energy. Assessing microbial activity in the natural atmosphere remains highly challenging, if not impossible. The development of methods able to detect and quantify microbial metabolic activity in air-suspended cells and at high frequency is therefore necessary to go further.

## 6 Data availability

Raw sequencing MGs and MTs data are available as fastq.gz files through the European Nucleotide Archive at EBI, under the
project accession PRJEB54740, samples ERR9966616 to ERR9966643.

## 7 Author contribution

Conceptualization: PA
Methodology: RP, FR, FE, BB, EN
Investigation: RP, PA
Visualization: RP, MJ, PA
Funding acquisition: PA, BE
Supervision: PA



Writing – original draft: RP, PA

Writing – review & editing: RP, PA, FR, MJ, FE, EN, BB, BE

**8 Competing interests**

The authors declare that they have no conflict of interest.

**9 Acknowledgments and funding**

We thank OPGC's facility for running the PUY atmospheric station and sharing meteorological data, and for logistical help during sampling. We are also grateful to Nadia Goué at the computational center of Clermont Auvergne University
(Mésocentre) and the network Auvergne Bioinformatique for providing computational power and excellent support with software deployment in the Galaxy environment. This research has been supported by the French National Research Agency (ANR) grant no. ANR-17-MPGA- 0013.




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
