# Peer review of "Clouds influence the functioning of airborne microorganisms"

_EGUsphere, 2024_

## Author Comment (AC1)

**Author response to comments by Referee #1.**

All referee comments are shown in black, our author responses in blue; suggested new manuscript text is indicated in red; text citations are *in italic*.

In the manuscript "Clouds influence the functioning of airborne microorganisms", by Péguilhan, et al., the authors explore the metabolic activity of cloud-borne microbes in comparison to airborne microbes using metagenomic and metatranscriptomic approaches. A thorough analysis has been conducted to explore possible mechanisms once microbes experiencing humid conditions in clouds compared to open air, with respect to stress response, metabolism, etc. However, some theories, such as the "birch effect," lack robust support, and I find the presented data unconvincing, as I specify below.

We thank the Referee for evaluating our work and for their suggestions to improve the manuscript. We provide a point-by-point response to the comments below.

Sample collection:

1- What were the negative controls for the cloud and air samples? This should be clarified in the methodology and presented as SI data.

We agree that this basic information regarding controls was missing, and we acknowledge for it. We will include the related following text in the Materials and Methods section about sample collection (Section 2.1):
"*Negative controls consisted of unexposed collection liquid, and of collection liquid exposed to the sampling tank for 10 minutes, sampler off. These were taken immediately before sampling, and processed in parallel of samples. For atmospheric samples,....*". and "*Samples and controls were processed immediately after sampling …*".
And, in Section 2.2 (Nucleic acid extraction and shotgun sequencing):
"*Only trace amounts of DNA could be obtained from negative controls (7.3 ng of DNA on average, 11.4 ng at maximum), and these were, thus, not processed for sequencing. In contrast, the total amounts of DNA and RNA recovered from environmental samples ranged from 42.6 to 838.7 ng and 22.5 to 244.8 ng, respectively. The corresponding total DNA and RNA concentrations in the air volumes sampled, as inferred from concentrations in the extracts, ranged from 0.03 to 0.73 ng DNA.m$^{-3}$ and from 0.026 to 0.42 ng RNA.m$^{-3}$, respectively (Table S1)*".
In addition, we detected a mistake related with conversion factors in the concentrations of DNA and RNA as reported per volume of air in Table S1, and this will be corrected. This does neither have impacts on the statistics (non-parametric) nor on the conclusions.

2- As the manuscript cannot include seasonality, authors should constrain their samples to a specific season. Specifically, the cloud sample during springtime could impact diversity and abundance and introduce seasonal-related impact.

We aimed at evaluating statistically the influence of a specific environmental variable on the functioning of airborne microorganisms, *i.e.* the presence of condensed water, which is expected to represent one of the major limitations for microbial activity in the atmosphere. We agree with the Referee that the lack of systematic data is a weakness of our study and does not

allow any conclusions about possible temporal variability on the functioning of airborne microorganisms, but such an aim is not the motivation of our study.

The diversity of microorganisms in the air is known to be highly variable over short temporal and spatial scales (Fierer et al., 2008). Every air sample differs from others in one or the other temporal, spatial, or environmental variables. We tried to minimize this variability by contrasting samples collected at a single location (puy de Dôme Mountain station), using unique sampling methods. This site is conveniently exposed to ranges of meteorological conditions including the alternance of clear conditions and clouds, and also to variations of temperature, humidity, position respect to the boundary layer, air mass origin, etc. At this site, clouds are inherently more frequent during the cold season than during summer (Baray et al., 2019). To follow our hypothesis that condensed water may affect microbial functioning, we categorized the samples among two categories discriminated by the presence, or not, of liquid water (*i.e.*, RH >100%, LWC >0). In our study, clear atmosphere was collected in July, September and November, and clouds in March, July, October and November, so at similar periods of the year except for the March sample. This one particular cloud sample was not an outlier in our dataset, so except for prejudices there is no specific rationale for excluding it. We will develop this in the concluding paragraph in Section 5, as:

 "*Our study focused in particular on the potential impact of clouds on microbial functioning, and it relies on samples collected on a single site, using unique sampling methods in order to avoid introducing site effects and methodological bias. We could qualitatively show that there are differences in microbial gene expressions in samples collected in cloud-free vs cloudy air masses. We used 'water availability' as a proxy to distinguish the two air mass types. However, cloudy air masses also differ from those outside clouds in a multitude of other environmental factors which are expected to play roles on aeromicrobiome's functioning, and they still need to be evaluated (Amato et al., 2023). Such variables include temperature, solar radiation, chemical composition, etc, and they are linked not only to clouds but also to altitude, location, day/night cycles and season. The synergy, temporal dynamics and arrangement of these variables (shocks, cloud cycles, freezing events, combination of chemicals, etc…) could also participate in shaping the aeromicrobiome in even more complex ways.*".

3- Table 1 - What was the time of sampling? It is not specified whether samples were collected during day or night.

All the samples were collected during daytime, typically from ~9:00 AM to 3:00 PM local time. This information was indeed missing, so we will add it in Table 1, and mention it in the Methods Section 2.1:
"*A total of nine cloud and six clear air events were sampled in 2019 and 2020, for periods of about two to six consecutive hours, during daytime.*".

Discussion:

4- P. 15, L. 56: I'm afraid authors are overstating their findings, suggesting this is the first time demonstrating the impact of atmospheric conditions on microbial functioning in the atmosphere (See Bryan et al, 2019, and others).

This comment relates to our statement that "*For the very first time, we demonstrate that atmospheric conditions influence multiple facets of microbial functioning in the natural atmosphere.*"
The study by (Bryan et al., 2019) did not report microbial functioning, but rather structural biodiversity from amplicons, and evidence for viability from cultures and ATP. General

indicators of microbial viability and potential metabolic activity such as cultures, ATP or RNA are frequently reported in atmospheric samples (Amato et al., 2007, 2017; Fahlgren et al., 2010; Hill et al., 2007; Šantl-Temkiv et al., 2018; Vaïtilingom et al., 2012; Wirgot et al., 2017), but they do not provide any information about the actual biological functions themselves. We therefore disagree with Referee's comment.

The impact of environmental conditions on the survival of aerosolized microorganisms has long been studied in atmospheric simulation chambers (Ehrlich et al., 1970; Wright et al., 1969). Similarly, the influence of environmental variables on the metabolic activity of airborne cells was actually first demonstrated in simulation chamber, where the presence of volatile organic compounds could be linked with elevated ribosome content in bacteria cells (Krumins et al., 2014). In the natural environment, (Wirgot et al., 2017) observed a positive correlation between the concentrations of $H_2O_2$ and ATP in clouds, suggesting some extent of causality on the metabolism, but no indication regarding the biological functions actually involved. To our knowledge, so far only (Amato et al., 2019) reported biological functioning of microorganisms in the natural atmosphere, without a priori (using untargeted methods), on samples captured into a fixative agent. This study did however not investigate possible relationships with environmental conditions, specifically the presence of clouds, which is the purpose of the present study.

Hence, unless we are missing an important reference, we decide to maintain our statement that our study is the first to bring evidence for natural atmospheric conditions influencing multiple airborne microbial biological functions, but we rephrase it as follows:

"*For the first time, we demonstrate that multiple microbial functions are directly influenced by atmospheric conditions, specifically by the presence of clouds*.".

5- Section 4.1: it is problematic to deduce from higher RNA:DNA levels that the metabolic levels in clouds are higher. Especially as the annotated genes in MT are not significantly different between the two environments, as seen in Table S1. Instead, it seems that the levels of DNA gene annotation in clouds are the factor that results in a higher RNA:DNA ratio in clouds.

We admit that our text may not have been fully clear. The amount of nucleic acids (in mass) and the number of annotated genes (in number of unique hits of the sequences against the database) are independent variables. The conclusion that clouds exhibit higher levels of microbial metabolic activity is based on the respective amounts of RNA and DNA extracted from samples, not from the numbers of annotated genes. The RNA-to-DNA concentration ratio in samples ranged from 0.21 to 3.62 overall, and this was significantly higher in clouds than in clear atmosphere, as shown in Fig 1A and Table S1. To avoid confusion, we will specify in the Discussion RNA-to-DNA "*concentration*" ratio:

"*Higher concentration of RNA, respect to DNA, in clouds than in cloud-free air suggests higher levels of microbial metabolic activity (Baldrian et al., 2012; Salazar et al., 2019). This remains to be assessed quantitatively through more direct activity measurements.*"

6- Thus, the Birch effect doesn't seem likely to explain your findings. Instead, I suggest considering an environmental switch of specific genes as related to the environmental conditions.

The Birch effect corresponds to a pulse of $CO_2$ emission from dry soils following rainfall, due to the promotion of microbial activity by water (Griffiths and Birch, 1961). We observed signs of increased biological activity in relation with liquid water. Hence this analogy appears sound with regards to what is known in soils. We did not evaluate it here, but the release of $CO_2$ from

biological activity in clouds is expected to be negligible compared to soils due to much lower biomass (Ervens and Amato, 2020; Vaïtilingom et al., 2013).

The current text in the manuscript clearly refers to, in particular, the increase of biological activity during the Birch effect, but we will make it clearer in the Discussion Section "*A 'Birch effect' up in the sky*", as:

"*The switch of functional gene expression observed in microorganisms between clear atmosphere and clouds therefore suggests a phenomenon similar to the "Birch effect" that occurs in dry soils in response to rewetting by rain, where the sudden influx of water triggers a burst of microbial activity*".

As it is stated in the current text, the impact in terms of gas fluxes is necessarily low: "*As with soils – but necessarily to a much smaller extent because of the low biomass – the resurgence of microbial activity in clouds may lead to the release of gaseous biogenic compounds […] which is also supported by the overrepresentation of ammonium utilization process*".

7- Moreover, if a birch effect occurs, I suspect it would be linked with spore-forming species, and the transformation from the dormant to the vegetative form would be characterized by key genes that should be presented to support the proposed theory.

Indeed, elements in support of spore germination are discussed in Section 4.2 "Airborne fungal spores initiate germination in clouds". We will extend this to include mentions of specific biomarkers overrepresented in cloud metatranscriptomes, as:

"*In agreement with the overrepresentation of numerous transcripts, it is likely that fungal spores initiate germination in clouds. These include translation initiation and elongation factors affiliated to several taxa of fungi (elF4E, eEF3 and others) (Osherov and May, 2001; Van Leeuwen et al., 2013; Li et al., 2022), chitin deacetylase (Leroch et al., 2013) and other regulatory protein genes such as areA (Kudla et al., 1990).*".

8- Section 4.3: This section appears to rely more on generalizations than on solid data. I recommend either omitting this part or revising it for clarity and support.

The possibility that microorganisms could multiply in atmospheric water (clouds, fog), supported by dissolved nutrients and liquid water, was suggested earlier from others (Fuzzi et al., 1997; Sattler et al., 2001). We agree that this section about biomass production is not sufficiently supported by data in our work, so we will merge this section with the next Discussion Section "*Utilization of nutrients and interactions with chemistry*", and modify the text accordingly as:

"*Microbial activity is driven by the balance between water availability and accessibility to substrates (Skopp et al., 1990). Although not evaluable here, the amounts of water retained by efflorescent aerosols below water vapor saturation may be sufficient to sustain microbial activity, down very low values of relative humidity (Cruz and Pandis, 2000; Ervens et al., 2024). In clouds, i.e., above saturation levels, the large amounts of available water make it even conceivable that bacterial multiplication occurs. Bulk cloud water indeed contains enough nutrients to sustain microbial growth including carboxylic acids, aldehydes, sugars, amino-acids, ammonium, nitrate, etc. (Amato et al., 2007; Bianco et al., 2016, 2018, 2019; Deguillaume et al., 2014; Renard et al., 2022), and the level of microbial activity at 0°C was shown to be compatible with it (Sattler et al., 2001). Field observations indicate that fog carries higher biomass than clear atmosphere (Fuzzi et al., 1997; Saikh and Das, 2023), while estimations suggest that microbial mass may double during cloud's lifetime (Ervens and Amato, 2020). The fact that statistically only 1 out of ~10 000 droplets contains a microbial cell in aerially suspended water, as opposed to bulk water, potentially causes a very efficient and rapid*

*depletion of nutrients in these small biotic volumes (Khaled et al., 2021) (~10$^{-6}$ µl for 20 µm diameter droplets, so a cell concentration of at least ~10$^9$ cells mL$^{-1}$ in biotic droplets), which exposes cells to starvation and may limit metabolic processes (Gray et al., 2019).*
*The overrepresentation of transcripts related to carbon, ammonium and nitrate utilization in clouds supports that carbon and nitrogen biological processing occurs ....".*

9- Figure 1A: Change "clear situation" to "clear atmosphere/open air". Also seen in Fig. 4 and across the manuscript.

We thank the Referee for pointing this out. Where necessary, "*clear situation*" will be modified as "*clear atmosphere*". The following figures will thus be updated: Fig 1, Fig 3, Fig 4, and Fig S3, S8, S9, S11, as well as Tables 1 and S1.

References cited:

Amato, P., Parazols, M., Sancelme, M., Mailhot, G., Laj, P., and Delort, A.-M.: An important oceanic source of micro-organisms for cloud water at the Puy de Dôme (France), Atmospheric Environment, 41, 8253–8263, https://doi.org/10.1016/j.atmosenv.2007.06.022, 2007.

Amato, P., Joly, M., Besaury, L., Oudart, A., Taib, N., Moné, A. I., Deguillaume, L., Delort, A.-M., and Debroas, D.: Active microorganisms thrive among extremely diverse communities in cloud water, PLOS ONE, 12, e0182869, https://doi.org/10.1371/journal.pone.0182869, 2017.

Amato, P., Besaury, L., Joly, M., Penaud, B., Deguillaume, L., and Delort, A.-M.: Metatranscriptomic exploration of microbial functioning in clouds, Scientific Reports, 9, 4383, https://doi.org/10.1038/s41598-019-41032-4, 2019.

Amato, P., Mathonat, F., Nuñez Lopez, L., Péguilhan, R., Bourhane, Z., Rossi, F., Vyskocil, J., Joly, M., and Ervens, B.: The aeromicrobiome: the selective and dynamic outer-layer of the Earth's microbiome, Frontiers in Microbiology, 14, 2023.

Baldrian, P., Kolařík, M., Stursová, M., Kopecký, J., Valášková, V., Větrovský, T., Zifčáková, L., Snajdr, J., Rídl, J., Vlček, C., and Voříšková, J.: Active and total microbial communities in forest soil are largely different and highly stratified during decomposition, ISME J, 6, 248–258, https://doi.org/10.1038/ismej.2011.95, 2012.

Baray, J.-L., Bah, A., Cacault, P., Sellegri, K., Pichon, J.-M., Deguillaume, L., Montoux, N., Noel, V., Seze, G., Gabarrot, F., Payen, G., and Duflot, V.: Cloud Occurrence Frequency at Puy de Dôme (France) Deduced from an Automatic Camera Image Analysis: Method, Validation, and Comparisons with Larger Scale Parameters, Atmosphere, 10, 808, https://doi.org/10.3390/atmos10120808, 2019.

Bianco, A., Voyard, G., Deguillaume, L., Mailhot, G., and Brigante, M.: Improving the characterization of dissolved organic carbon in cloud water: Amino acids and their impact on the oxidant capacity, Scientific Reports, 6, https://doi.org/10.1038/srep37420, 2016.

Bianco, A., Deguillaume, L., Vaïtilingom, M., Nicol, E., Baray, J.-L., Chaumerliac, N., and Bridoux, M.: Molecular Characterization of Cloud Water Samples Collected at the Puy de Dôme (France) by Fourier Transform Ion Cyclotron Resonance Mass Spectrometry, Environ. Sci. Technol., 52, 10275–10285, https://doi.org/10.1021/acs.est.8b01964, 2018.

Bianco, A., Deguillaume, L., Chaumerliac, N., Vaïtilingom, M., Wang, M., Delort, A.-M., and Bridoux, M. C.: Effect of endogenous microbiota on the molecular composition of cloud water: a study by Fourier-transform ion cyclotron resonance mass spectrometry (FT-ICR MS), Scientific Reports, 9, 7663, https://doi.org/10.1038/s41598-019-44149-8, 2019.

Bryan, N. C., Christner, B. C., Guzik, T. G., Granger, D. J., and Stewart, M. F.: Abundance and survival of microbial aerosols in the troposphere and stratosphere, ISME J, 13, 2789–2799, https://doi.org/10.1038/s41396-019-0474-0, 2019.

Cruz, C. N. and Pandis, S. N.: Deliquescence and Hygroscopic Growth of Mixed Inorganic−Organic Atmospheric Aerosol, Environ. Sci. Technol., 34, 4313–4319, https://doi.org/10.1021/es9907109, 2000.

Deguillaume, L., Charbouillot, T., Joly, M., Vaïtilingom, M., Parazols, M., Marinoni, A., Amato, P., Delort, A.-M., Vinatier, V., Flossmann, A., Chaumerliac, N., Pichon, J. M., Houdier, S., Laj, P., Sellegri, K., Colomb, A., Brigante, M., and Mailhot, G.: Classification of clouds sampled at the puy de Dôme (France) based on 10 yr of monitoring of their physicochemical properties, Atmos. Chem. Phys., 14, 1485–1506, https://doi.org/10.5194/acp-14-1485-2014, 2014.

Ehrlich, R., Miller, S., and Walker, R. L.: Relationship Between Atmospheric Temperature and Survival of Airborne Bacteria, Appl Microbiol, 19, 245–249, 1970.

Ervens, B. and Amato, P.: The global impact of bacterial processes on carbon mass, Atmospheric Chemistry and Physics, 20, 1777–1794, https://doi.org/10.5194/acp-20-1777-2020, 2020.

Ervens, B., Amato, P., Aregahegn, K., Joly, M., Khaled, A., Labed-Veydert, T., Mathonat, F., Nuñez López, L., Péguilhan, R., and Zhang, M.: Ideas and perspectives: Microorganisms in the air through the lenses of atmospheric chemistry and microphysics, EGUsphere, 1–18, https://doi.org/10.5194/egusphere-2024-2377, 2024.

Fahlgren, C., Hagström, Å., Nilsson, D., and Zweifel, U. L.: Annual Variations in the Diversity, Viability, and Origin of Airborne Bacteria, Appl. Environ. Microbiol., 76, 3015–3025, https://doi.org/10.1128/AEM.02092-09, 2010.

[revised manuscript text omitted]

---

## Author Comment (AC2)

**Author response to comments by Referee #2.**

All referee comments are shown in black, our author responses in blue; suggested new manuscript text is indicated in red; text citations are *in italic*.

This study by Péguilhan et al. investigates microbial activity in clouds, comparing it to samples from clear atmospheric conditions using metatranscriptomic and metagenomic sequencing. The results revealed a higher RNA-to-DNA ratio in cloud samples than in clear atmosphere samples, indicating elevated microbial metabolic activity. Metabolic pathways associated with various cellular processes were found to be overexpressed in cloud samples. The authors attributed this increased metabolic activity to the availability of moisture in clouds, which is absent under clear conditions. Despite the limited number of samples analyzed, the study is significant, as collecting samples for metagenomic and metatranscriptomic analyses is not trivial due to the low biomass in the atmosphere. This research provides valuable groundwork for future studies in this area.

We thank the referee for their positive assessment of our manuscript and for the constructive comments that lead to several clarifications and improvements.

Major comments:

1.    Sections 3.2.1 to 3.2.3 are primarily descriptive, listing overexpressed functions. To enhance clarity and strengthen the data presentation, the authors should consider structuring the discussion around specific research questions or hypotheses. This would create a more cohesive narrative, allowing the data to directly address these questions or test the proposed hypotheses.

These sections are indeed purely descriptive and factual, as they report results. We propose to better frame these without deeply modifying the whole structure of the manuscript, by adding some elements of discussion in the Result section:
"*These observations concur with increased biochemical energy needs in clouds.*". (Section 3.2.1).
"*The overrepresentation in clouds of transcripts of the regulatory gene areA suggests that multiple nitrogen sources are targeted (Kudla et al., 1990), likely as a response to limited resources. Clouds are also associated with aminoacid starvation (GO:0034198).*". (Section 3.2.2).
"*Such functional patterns can be interpreted as microbial responses to wetting. They indicate a probable sheltering effect of condensed water against oxidative stress, along with limited nutrient resources requiring metabolic adjustments.*". (Section 3.2.3).

2.    Related to the comment above, the Introduction could more clearly articulate the research questions the study aims to address. Rather than simply determining whether microbes are active and expressing genes in clouds, the authors should frame the study around more focused, in-depth questions.

We thank the Referee for this relevant suggestion. We clarify our objectives by adding the following text in the introduction:
"*Here, we postulate that clouds could act as atmospheric "oases", i.e., specific volumes providing water and nutrients to living organisms and allowing them to thrive within an*

*otherwise vast and hostile atmospheric environment. By using an innovative combined non-targeted metagenomics/metatranscriptomics approach, we examine the functioning of airborne microbial cells in clouds as compared with clear atmosphere, and specify if and which biological processes are indeed affected. Given that airborne particles, including bacteria, spend on average 10 – 15% of their atmospheric residence time in clouds (Ervens and Amato, 2020; Lelieveld and Crutzen, 1990), such oases would provide conditions of (temporary) habitats or 'airborne ecosystems' and therefore could lead to enhanced survival, persistence and dispersal of bacteria similar to features of other dynamic environments. This study, based on unique and unprecedented data sets, provides valuable information regarding the active aeromicrobiome and its environmental drivers.".*

3.  Section 2.3. Please provide a more detailed explanation of how the metagenomic and metatranscriptomic data were normalized. Additionally, it appears the authors analyzed short reads for this study. Did they attempt to assemble these reads into contigs or even reconstruct genomes?

Short reads were assembled in contigs in order to predict the genes and construct a gene catalog, as specified in the main text, Section 2.3:
"*This was elaborated by (I) merging all the **contigs** from each individual MG, (II) predicting genes...*", and in the Supplementary Materials and Figure S1: "*Each individual dataset of non-RNA reads in MGs (each sample) was first de novo assembled using MEGAHIT (v 1.1.3.5) (Li et al., 2015), with default parameters and a minimum contig length of 500 bp.*".
Short reads were then mapped against the gene catalog to obtain a coverage for DNA and RNA data (MG and MT, respectively), and RNA coverage was normalized to that of DNA using the MTXmodel R package, as specified:
"*Finally, (iv) non-rRNA reads in each MG and MT were mapped toward the annotated gene catalog*", and in supplementary material "*Non-rRNA gene sequences from all MGs and MTs were finally mapped to the gene catalog to obtain read counts per gene using BWA-MEM (v 0.7.17.1) (Li and Durbin, 2009) with default parameters.*" and "*Data normalization and differential expression analysis (DEA) were performed using the R package MTXmodel (R v4.0.3; MTXmodel v1.5.1) (Zhang et al., 2021).*".
For more clarity on data normalization, we will extend the Material and Methods section as:
"*In addition, statistical differential expression analysis (DEA) was performed on the MT to MG mapping coverages ratio towards the gene catalog in order to detect overrepresented genes and functions, and those significantly overrepresented in clouds compared to clear conditions, or conversely [MTX model v1.5.1 (Zhang et al., 2021); see supplementary material for details].*"
We could reconstruct contigs of up to 200,000 kb from MGs, but no complete genome. We recognize that metagenomes-assembled genomes (MAGs) are powerful tools to examine microbial diversity at deep taxonomic level, and investigate genomes organization, but these were not in the scope on the study. Given the low biomass and high diversity of airborne microbes, the recovery of sufficient quantities of high-quality DNA from atmospheric samples remains a challenge.

4.  Given the low biomass of the samples, please describe the procedures implemented to prevent contamination during sampling. Were negative controls used, and were any decontamination procedures applied to the sequencing reads?

The atmosphere is indeed one of the most dilute environments on Earth in terms of biomass (e.g., (Ervens et al., 2024; Šantl-Temkiv et al., 2022)). Great care should therefore be taken to

prevent contaminations. Basic precautions were taken during sampling such as clearing the area around samplers (located on a platform on the roof of puy de Dôme station), limiting human activity around samplers, along with practices including the systematic use of sterile material and UV decontaminated laminar flow hoods to handle samples, etc. All the solutions used were filtered through 0.2µm porosity and sterilized before use.

Negative controls consisted of unexposed collection liquid, and collection liquid exposed to the sampling tank for 10 minutes. All the details concerning negative controls are indicated in the supplementary material. However, we recognize that information regarding controls was missing in the main text, and we will therefore include some more explanations as described in the Response to the comment 1 of Referee #1.

Response to Referee #1, comment 1:

[We agree that this basic information regarding controls was missing, and we acknowledge for it. We propose to include the related following text in the Materials and Methods section about sample collection (Section 2.1):

"*Negative controls consisted of unexposed collection liquid, and of collection liquid exposed to the sampling tank for 10 minutes, sampler off. These were taken immediately before sampling, and processed in parallel of samples. For atmospheric samples, ....*". and "*Samples and controls were processed immediately after sampling …*".

And, in Section 2.2 (Nucleic acid extraction and shotgun sequencing):

"*Only trace amounts of DNA could be obtained from negative controls (7.3 ng of DNA on average, 11.4 ng at maximum), and these were, thus, not processed for sequencing. In contrast, the total amounts of DNA and RNA recovered from environmental samples ranged from 42.6 to 838.7 ng and 22.5 to 244.8 ng, respectively. The corresponding total DNA and RNA concentrations in the air volumes sampled, as inferred from concentrations in the extracts, ranged from 0.03 to 0.73 ng DNA.m$^{-3}$ and from 0.026 to 0.42 ng RNA.m$^{-3}$, respectively (Table S1)*".

In addition, we detected a mistake related with conversion factors in the concentrations of DNA and RNA as reported per volume of air in Table S1, and this will be corrected. This does neither have impacts on the statistics (non-parametric) nor on the conclusions.]

The decontamination of the sequencing reads consisted of the removal of human, embryophytes and metazoan reads, to focus on microbial sequences. This is specified in the supplement as: "*Human reads were filtered from the non-rRNA gene reads using Bowtie2 (v 2.4.2) (Langmead and Salzberg, 2012), against the NCBI Homo sapiens genome "hg38_2021-5-18" with default parameters (Tables S2-S3). Human reads were excluded from further analyses.*" And "*Only genes with >10 mapped sequences in MGs were considered, and the count tables for MGs and MTs were filtered in order to remove genes affiliated with "Embryophytes" and "Metazoa" and focus on microbial genes.*"

5.  Section 3.1. Currently, there is no figure on taxonomy in the main manuscript. Including a figure in the main text, rather than keeping all of them in the supplementary information, would improve readability and benefit the readers.

We thank the reviewer for this useful comment. We will combine the panels A and C of Fig S4 and S5 and present them in the main text as the new Fig 1 as shown below. The Alpha diversity indexes from Fig S4 and S5 will be combined in a new Fig S4, also shown below.

[Figure]

Figure 1. Bacterial and eukaryotic diversity from metagenomes. (A, B) Distribution of the most abundant bacterial and eukaryotic orders in the metagenomes, and corresponding hierarchical clustering (Ward's method, "ward.D2"). The intensity scale depicts centered-log ratio (clr) abundance. EnvType: environment type (the samples are identified as follows: "A" for clear atmosphere (air) or "C" for clouds, followed by the sampling date in the format "mmdd"); (C, D) Venn diagrams depicting the distribution of bacteria and eukaryotic genera between clouds and clear atmosphere.

[Figure]

Fig S4.
Alpha diversity indexes (observed and estimated richness, Shannon's diversity and Inverse Simpson's evenness) in clear atmosphere and cloud metagenomes at the genus level for (A) bacteria and (B) eukaryotes.

6. Section 3.2.3. Several stress-related pathways are described in this section, but they are not further elaborated in the Discussion. Including a brief discussion on stress tolerance would help readers understand the challenges microbes face and how they adapt to them.

We thank the Reviewer for this comment. We agree that stress is an important aspect regarding the aeromicrobiome, and we will add the following new section in the discussion about stress responses, which attest of multiple metabolic regulations.

*"4.2 Responses to stress attest of multiple functional adjustments*

*Our data indicate that clear atmosphere is dominated by responses to oxidative stress and DNA damages, involving SOS response, while clouds are characterized by osmotic stress, starvation and autophagy. The functional patterns of aeromicrobiome' stress responses are therefore very consistent with environmental conditions, and help drawing a more complete picture of the multiple aspects of the microbial journey in the high atmosphere.*

*In clouds, liquid water shelters cells against oxidants and radiations, but the rapid condensation/evaporation processes along with the dissolution of solids and the solubilization of gases generate large fluctuations of water activity (e.g., (Koehler et al., 2006)). Additionally, in the limited volumes provided by droplets, the nutrient requirements may often not be fully satisfied, and autophagy processes may contribute to alleviating the needs. Peroxisomes, organelles dedicated to the detoxification of oxidants in eukaryotes, are targeted in particular by autophagy (pexophagy), as during fungal spore germination. Such process could compromise survival if the cloud evaporates, but it may be a trade-off with increased chances of success in the race for surface colonization if the cloud precipitates.*

*Here, clear air was collected at relative humidity between 41%-78%, i.e., at the limits of compatibility with biological processes, around ~0.6 aw (water activity) for the most tolerant organisms (i.e., 60% pure water rH) (Stevenson et al., 2015). At aw below 0.55, DNA gets unstructured and metabolic regulations are no longer possible. Water limitation is a great challenge that many microorganisms have to face in their natural habitats. This affects cell turgor due to water efflux and slows down growth and metabolic activity (Chowdhury et al., 2011).*

*In order to manage the numerous environmental factors related with variations of water activity, such as temperature or osmotic pressure, microorganisms have developed ranges of strategies: modifications of the saturation level of lipids in membranes to adjust fluidity, synthesis and accumulation of intracellular compatible solutes in order to prevent water efflux and maintain homeostasis (osmoprotectants and cryoprotectants such as $K^+$, sucrose, trehalose, amino-acids and others)* (Poolman and Glaasker, 1998)*, chaperones to protect molecular structures, membrane canal proteins, such as aquaporins, to sustain water fluxes* (Tong et al., 2019)*, etc.".*

7. Section 4.3. The authors suggest that microbial growth may occur in clouds. Were any genes related to cell replication overexpressed in the cloud samples?

This section will no longer be maintained in the manuscript and will be merged with the Section "*Utilization of nutrients and interactions with chemistry*". (See response to comment 8 by Referee #1).
Response to Referee #1, comment 8:
[The possibility that microorganisms could multiply in atmospheric water (clouds, fog), supported by dissolved nutrients and liquid water, was suggested earlier from others (Fuzzi et al., 1997; Sattler et al., 2001). We agree that this section about biomass production is not sufficiently supported by data in our work, so we will merge this section with the next section about "Utilization of nutrients and interactions with chemistry", and modify the text accordingly as:
"*Microbial activity is driven by the balance between water availability and accessibility to substrates (Skopp et al., 1990). Although not evaluable here, the amounts of water retained by*

*efflorescent aerosols below water vapor saturation may be sufficient to sustain microbial activity, down very low values of relative humidity (Cruz and Pandis, 2000; Ervens et al., 2024). In clouds, i.e., above saturation levels, the large amounts of available water make it even conceivable that bacterial multiplication occurs. Bulk cloud water indeed contains enough nutrients to sustain microbial growth including carboxylic acids, aldehydes, sugars, amino-acids, ammonium, nitrate, etc. (Amato et al., 2007a; Bianco et al., 2016, 2018, 2019; Deguillaume et al., 2014; Renard et al., 2022), and the level of microbial activity at 0°C was shown to be compatible with it (Sattler et al., 2001). Field observations indicate that fog carries higher biomass than clear atmosphere (Fuzzi et al., 1997; Saikh and Das, 2023), while estimations suggest that microbial mass may double during cloud's lifetime (Ervens and Amato, 2020). The fact that statistically only 1 out of ~10 000 droplets contains a microbial cell in aerially suspended water, as opposed to bulk water, potentially causes a very efficient and rapid depletion of nutrients in these small biotic volumes (Khaled et al., 2021) (~$10^{-6}$ µl for 20 µm diameter droplets, so a cell concentration of at least ~$10^9$ cells mL$^{-1}$ in biotic droplets), which exposes cells to starvation and may limit metabolic processes (Gray et al., 2019).*
*The overrepresentation of transcripts related to carbon, ammonium and nitrate utilization in clouds supports that carbon and nitrogen biological processing occurs ....".]*

In addition, numerous overrepresented transcripts relate to translation initiation and elongation factors in clouds. These likely indicate fungal spore germination. Some text will be added in the Results Section "*3.2.1 Central, carbon and energy metabolisms*" as:
"*Numerous transcripts related to translation and elongation factors in Eukaryotes are overrepresented in clouds (Data S6) suggesting metabolic regulations and the production of new biomass.*",
and in the Discussion section "*Airborne fungal spores initiate germination in clouds*" as:
 "*In agreement with numerous overrepresented transcripts, it is likely that fungal spores initiate germination in clouds. These included translation initiation and elongation factors affiliated with several taxa of fungi (elF4E, eEF3 and others) (van Leeuwen et al., 2013; Li et al., 2022; Osherov and May, 2001), chitin deacetylase (Leroch et al., 2013) and other regulatory protein genes such as area (Kudla et al., 1990).*".

8.    A brief discussion on the limitations of this study is necessary to put the findings into perspective.

Limitations of the study are discussed in the concluding section. As recommended, we will extend the Discussion, in particular with 2 paragraphs about (i) the limitations of metatranscriptomics to quantitatively evaluate microbial activity, and (ii) other environmental variables than clouds that may contribute to variations in aeromicrobiome's functioning.
"*Transcriptomes attest of potential cellular activity, but they do not provide quantitative information of microbial activity in terms of fluxes of elements or energy. Quantitative measurements of microbial activity therefore remain necessary to confirm the "atmospheric Birch effect" caused by clouds. The transitions between clear and cloudy conditions in particular remain to be examined to evaluate the temporal responsiveness of airborne microbial assemblages to cloud formation and evaporation. While this is potentially achievable in an atmospheric simulation chamber, assessing microbial activity in naturally aerially suspended biological microorganisms remains highly challenging, if not impossible (yet). The development of methods able to detect and quantify microbial metabolic activity in air-suspended cells and at high frequency appears therefore as a prerequisite.*
*Our study focused in particular on the potential impact of clouds on microbial functioning, and it relies on samples collected on a single site, using unique sampling methods in order to avoid*

*introducing site effects and methodological bias. We could qualitatively show that there are differences in microbial gene expressions in samples collected in cloud-free vs cloudy air masses. We used liquid water content as a proxy to distinguish the two air mass types. However, cloudy air masses also differ from those outside clouds in a multitude of other environmental factors which are expected to play roles on aeromicrobiome's functioning, and they still need to be evaluated (Amato et al., 2023). Such variables include temperature, solar radiation, chemical composition, etc, and they are linked not only to clouds but also to altitude, location, day/night cycles and season. The synergy, temporal dynamics and arrangement of these variables (shocks, cloud cycles, freezing events, combination of chemicals, etc...) could also participate in shaping the aeromicrobiome in even more complex ways.*".

References cited:

Amato, P., Mathonat, F., Nuñez Lopez, L., Péguilhan, R., Bourhane, Z., Rossi, F., Vyskocil, J., Joly, M., and Ervens, B.: The aeromicrobiome: the selective and dynamic outer-layer of the Earth's microbiome, Frontiers in Microbiology, 14, 2023.

Chowdhury, N., Marschner, P., and Burns, R.: Response of microbial activity and community structure to decreasing soil osmotic and matric potential, Plant Soil, 344, 241–254, https://doi.org/10.1007/s11104-011-0743-9, 2011.

Ervens, B. and Amato, P.: The global impact of bacterial processes on carbon mass, Atmospheric Chemistry and Physics, 20, 1777–1794, https://doi.org/10.5194/acp-20-1777-2020, 2020.

Ervens, B., Amato, P., Aregahegn, K., Joly, M., Khaled, A., Labed-Veydert, T., Mathonat, F., Nuñez López, L., Péguilhan, R., and Zhang, M.: Ideas and perspectives: Microorganisms in the air through the lenses of atmospheric chemistry and microphysics, EGUsphere, 1–18, https://doi.org/10.5194/egusphere-2024-2377, 2024.

Koehler, K. A., Kreidenweis, S. M., DeMott, P. J., Prenni, A. J., Carrico, C. M., Ervens, B., and Feingold, G.: Water activity and activation diameters from hygroscopicity data - Part II: Application to organic species, Atmospheric Chemistry and Physics, 6, 795–809, https://doi.org/10.5194/acp-6-795-2006, 2006.

Kudla, B., Caddick, M. X., Langdon, T., Martinez-Rossi, N. M., Bennett, C. F., Sibley, S., Davies, R. W., and H N Arst, J.: The regulatory gene areA mediating nitrogen metabolite repression in Aspergillus nidulans. Mutations affecting specificity of gene activation alter a loop residue of a putative zinc finger, The EMBO Journal, 9, 1355, https://doi.org/10.1002/j.1460-2075.1990.tb08250.x, 1990.

Langmead, B. and Salzberg, S. L.: Fast gapped-read alignment with Bowtie 2, Nat Methods, 9, 357–359, https://doi.org/10.1038/nmeth.1923, 2012.

van Leeuwen, M. R., Krijgsheld, P., Bleichrodt, R., Menke, H., Stam, H., Stark, J., Wösten, H. A. B., and Dijksterhuis, J.: Germination of conidia of Aspergillus niger is accompanied by major changes in RNA profiles, Studies in Mycology, 74, 59–70, https://doi.org/10.3114/sim0009, 2013.

Lelieveld, J. and Crutzen, P. J.: Influences of cloud photochemical processes on tropospheric ozone, Nature, 343, 227–233, https://doi.org/10.1038/343227a0, 1990.

Leroch, M., Kleber, A., Silva, E., Coenen, T., Koppenhöfer, D., Shmaryahu, A., Valenzuela, P. D. T., and Hahn, M.: Transcriptome Profiling of Botrytis cinerea Conidial Germination Reveals Upregulation of Infection-Related Genes during the Prepenetration Stage, Eukaryotic Cell, 12, 614–626, https://doi.org/10.1128/ec.00295-12, 2013.

Li, C., Jia, S., Rajput, S. A., Qi, D., and Wang, S.: Transcriptional Stages of Conidia Germination and Associated Genes in Aspergillus flavus: An Essential Role for Redox Genes, Toxins, 14, 560, https://doi.org/10.3390/toxins14080560, 2022.

Li, D., Liu, C.-M., Luo, R., Sadakane, K., and Lam, T.-W.: MEGAHIT: an ultra-fast single-node solution for large and complex metagenomics assembly via succinct de Bruijn graph, Bioinformatics, 31, 1674–1676, https://doi.org/10.1093/bioinformatics/btv033, 2015.

Li, H. and Durbin, R.: Fast and accurate short read alignment with Burrows–Wheeler transform, Bioinformatics, 25, 1754–1760, https://doi.org/10.1093/bioinformatics/btp324, 2009.

Osherov, N. and May, G. S.: The molecular mechanisms of conidial germination, FEMS Microbiol Lett, 199, 153–160, https://doi.org/10.1111/j.1574-6968.2001.tb10667.x, 2001.

Poolman, B. and Glaasker, E.: Regulation of compatible solute accumulation in bacteria, Molecular Microbiology, 29, 397–407, https://doi.org/10.1046/j.1365-2958.1998.00875.x, 1998.

Šantl-Temkiv, T., Amato, P., Casamayor, E. O., Lee, P. K. H., and Pointing, S. B.: Microbial ecology of the atmosphere, FEMS Microbiology Reviews, fuac009, https://doi.org/10.1093/femsre/fuac009, 2022.

Stevenson, A., Burkhardt, J., Cockell, C. S., Cray, J. A., Dijksterhuis, J., Fox-Powell, M., Kee, T. P., Kminek, G., McGenity, T. J., Timmis, K. N., Timson, D. J., Voytek, M. A., Westall, F., Yakimov, M. M., and Hallsworth, J. E.: Multiplication of microbes below 0.690 water activity: implications for terrestrial and extraterrestrial life, Environmental Microbiology, 17, 257–277, https://doi.org/10.1111/1462-2920.12598, 2015.

Tong, H., Hu, Q., Zhu, L., and Dong, X.: Prokaryotic Aquaporins, Cells, 8, 1316, https://doi.org/10.3390/cells8111316, 2019.

Zhang, Y., Thompson, K. N., Huttenhower, C., and Franzosa, E. A.: Statistical approaches for differential expression analysis in metatranscriptomics, Bioinformatics, 37, i34–i41, https://doi.org/10.1093/bioinformatics/btab327, 2021.